# Can Citrus Fiber Improve the Quality of Gluten-Free Breads?

**DOI:** 10.3390/foods12071357

**Published:** 2023-03-23

**Authors:** Raquel Bugarín, Manuel Gómez

**Affiliations:** Food Technology Area, College of Agricultural Engineering, University of Valladolid, 34004 Palencia, Spain

**Keywords:** gluten-free bread, citrus fiber, hydration, shear, texture

## Abstract

Citrus fiber has a high water absorption capacity, and its properties can be modified by shearing. In this study, the influence of the addition of normal or shear-activated citrus fiber was analyzed in two gluten-free bread formulations. Citrus fiber increases bread optimal hydration and breadcrumb alveolus size due to this high water retention capacity. However, results are negative in the formula based on starches and rice flour because specific volume is significantly reduced, while bread quality improves in the formula based on starches (corn and tapioca). In this case, the breads become less hard and more cohesive, elastic, and resilient, reducing staling. Baking yield also increased due to a greater hydration and a reduced weight loss during baking, without losing acceptability. The mechanical pre-activation of the fiber further increases optimal hydration, without major changes in the quality of the final bread. These effects are associated with cell rupture, and thus the formation of a three-dimensional network, including the increase of surface area and its interaction with water. Citrus fiber increases the hydration of the dough, as well as the cohesiveness, resilience, and elasticity of the crumb, reducing the increase in hardness during storage without affecting acceptability or increasing it.

## 1. Introduction

Bread is an essential food item for a large part of the world’s population. It is usually made with wheat flour, as the proteins present in this cereal have the unique ability to form, when hydrated and mechanically worked (kneading), a network (gluten network) that gives doughs extensibility and the ability to retain the air formed during fermentation [1]. However, some people cannot eat wheat-based or gluten-containing products due to intolerance, a wheat allergy, or coeliac disease [2].

To make gluten-free breads, starches, and flours from cereals, pseudo-cereals and other gluten-free grains are used; in these kinds of bread, starch plays an essential role [3]. The use of “gluten substitutes” is common in these preparations. Among these, hydroxypropyl methylcellulose (HPMC) gives rise to the best results in terms of bread volume [4]. However, products with HPMC have an excessively dry texture, which is harder than that of breads with a similar volume obtained with other hydrocolloids [5]. To compensate for this defect, it is common to mix HPMC with other hydrocolloids with greater water absorption capacity, such as psyllium, xanthan gum, or guar gum [6]. However, in the case of adding this type of hydrocolloids, it is necessary to modify dough hydration [7].

In gluten-free breads elaboration, the correct hydration of the dough is essential, since the greater the hydration—and therefore the less viscous the dough—the greater the expansion of the pieces, until they reach an excessive hydration that makes the dough fall during fermentation or baking [8,9]. Unlike wheat flour breads, in gluten-free breads there is no universally accepted equipment to analyze the hydration of the dough and modify it based on changes in the recipe, such as the farinograph. That is why it is interesting to know how breads change depending on the hydration of the doughs. 

Citrus fiber (CF)—a product of the circular economy [10] mainly composed of cellulose and pectins—is obtained from the waste of the citrus processing industry, mainly from the peel. This type of fiber, as well as some gums used in the production of gluten-free bread, has a high water absorption capacity [11,12]. It also brings interesting emulsifying properties [13]. Citrus fiber reincorporation has been studied mainly in meat products [14], but in baked products it is only limited to studies carried out by Miller [15] and Spina et al. [16] on wheat breads. In the case of gluten-free breads, Ozturk and Mert [17] compared CF functionality with that of xanthan gum, but their results showed that it did not optimize dough hydration. Korus et al. [18] proposed up to 20% CF incorporation to replace starches, in formulas that included other hydrocolloids such as pectin and guar gum that produced good results in texture and shelf life. In this case, they did optimize hydration, based on a dough texture test.

Some fibers from plant residues change their properties when subjected to shear forces, such as the passage through a high-pressure homogenizer, some types of mills, or ultraturrax [19,20]. In the case of citrus fiber, changes in its properties have also been observed after high-pressure homogenization, and/or ball mill [21,22]. These treatments in general lead to an increase in fiber hydration properties, which could also affect the optimal hydration of the gluten-free bread doughs containing them.

This paper analyses how the addition of citrus fiber (6%), without or after shear forces activation, modifies the properties of gluten-free starch and flour mixtures (hydration, gel, and pasting properties). Then, the possibility of incorporating CF (with or without modification) into the elaboration of gluten-free breads is studied. To this end, two basic formulas were used: a mixture of maize starches and tapioca (Control 1), and another based on these starches and rice flour (Control 2), to which citrus fiber was added both without prior activation (CF) and after mechanical activation (ACF). The influence of dough hydration on specific volume and the influence of fiber on weight loss, texture, and acceptability of the final product was analyzed.

## 2. Materials and Methods

### 2.1. Materials

Gluten-free breads were made with rice flour (Fraga S.A., Medina del Campo, Spain), an average particle size of 136 microns, maize starch (Tereos, Syral Iberia SAU, Zaragoza, Spain), and cassava starch (Yoki Alimentos SA, Paraná, Brazil). The rest of the ingredients used were refined sunflower oil (Langosta SA, Daimiel, Spain), sugar (Acor, Valladolid, Spain), instant dry baker’s yeast (Dosu Maya Mayacilik A.S, Istambul, Turkey), hydroxypropyl methylcellulose K4M (Rettenmaier Ibérica, Barcelona, Spain), Citrus Fiber Vitacel CF 312 (Rettenmaier Iberica, Barcelona, Spain), salt (Esco European Salt Company, Niedersachsen, Germany), and water from the local water supply.

To activate the citrus fiber, it was mixed with the water used for measurements and elaborations, and sheared in an ultraturrax (IKADispersor ultra-turrax T 10 Basic) at 2000 rpm for 1 min.

### 2.2. Methods

#### 2.2.1. Flour Characteristics (Hydration, Gel, and Pasting Properties)

Water-binding capacity (WBC) was determined using the AACC method 56-30.01 [23] with modifications, since samples with CF absorbed all the water after centrifugation. One point five grams (±0.1 g) of blends (rice flour, starches, and CF in the proportions shown in Table 1) were mixed with 25 mL of distilled water in centrifuge tubes. After vortex homogenization (MS2 Minishaker, IKA, Staufen, Alemania), the mixture was centrifuged at 2000× *g* for 15 min and the supernatant was removed. WBC was calculated as the amount of water retained per gram of dry sample.

The viscosity profile of the different samples was analyzed using a Rapid Visco Analyser (model RVA-4C, Newport Scientific Pty. Ltd., Warriewood, Australia), following the AACC method 76-21.02 [24].

Gels obtained from RVA analyses were kept under refrigeration at 4 °C for 24 h. Before texture assay, gels were tempered for 20 min at room temperature. Gel hardness was measured by a texturometer (model TA.XT2i Texture Analyzer, from Stable Micro Systems Ltd., Surrey, UK), which sets a compression cycle. The texturometer was equipped with a 5 kg load and a 50 mm diameter cylindrical probe, and calibrated with a return distance of 50 mm, return velocity of 20 mm/s, and 5 g contact force. A force–time curve was obtained, and the Positive Peak Force (N) data were selected [25].

All the samples were measured by triplicate.

#### 2.2.2. Bread Elaboration

Bread formulations, based on the work of Sigüenza-Andrés et al. [26], are shown in Table 1. All ingredients, except dry yeast and tap water, were mixed at speed 1 for 1 min using a KitchenAid Professional mixer (Kitchen Aid, St. Joseph, MI, USA) with a dough hook (K45DH). The yeast was mixed previously with water for its rehydration; it was mixed with the rest of the ingredients at speed 2 for 8 min. One hundred- and fifty-gram portions of bread dough were placed into oil-coated aluminum pans (159 × 109 × 39 mm) and fermented at 30 °C and 90% RH for 60 min. Doughs were baked at 190 °C in an oven (Salva, Lezo, Spain) for 40 min after fermentation. The aluminum pans were removed; the bread was left for 60 min so that it could reach room temperature, and was then placed in polyethylene bags. To calculate the optimal hydration, breads with increasing hydration were elaborated (increasing by 10% in each preparation), and the specific volume was analyzed after 24 h. An increase in the specific volume was observed as hydration increased until it caused the fall of the dough during fermentation or baking. Optimum hydration was considered that which generated breads with the highest specific volume. The curves obtained can be seen in Figure 1.

Breads elaborated with optimal hydration were stored at 22 °C for 7 days. They were analyzed 24 h and 7 days after elaboration. All the bread elaborations were performed twice.

#### 2.2.3. Evaluation of Dough Rheology

The rheological behavior of bread doughs with optimal hydration without yeast was studied after a 2-min rest using a controlled strain rheometer Thermo Scientific Haake RheoStress (Thermo Fisher Scientific, Schwerte, Germany). The constant temperature (25 °C) was controlled by a Phoenix II P1-C25P water bath (Thermo Fisher Scientific, Schwerte, Germany), with 60 mm diameter rough-surfaced titanium plates placed in parallel (PP60Ti). First, a strain sweep test was performed with a strain range of 0.1 to 100 Pa and a constant frequency of 1 Hz, to identify the linear viscoelastic region. With the strain sweep, the linear viscoelastic region was determined, and a stress value (т) was identified where G′ and G″ remained constant (slope 0). These data were used in the frequency sweep, from which the values of elastic modulus (G′ [Pa]), viscous modulus (G″ [Pa]), and tan δ were obtained based on frequency values (ω [Hz]). Values of G′ and G′′ at 1 Hz were selected. Each dough was analyzed in duplicate.

#### 2.2.4. Bread Characteristics

Bread characteristics were evaluated 24 h after baking, except for weight loss (after 1 h) and texture parameters (after 24 h and 7 days). 

The weight loss of bread during baking was determined in five pieces of each bread batch. This parameter was calculated using the following formula:$$Weight\;loss=\frac{dough\;weight-bread\;weight\;after\;baking}{dough\;weight}\times 100$$

Bread volume was measured in the same five pieces of bread of each elaboration using a Volscan Profiler volume analyzer (Stable Microsystems, Surrey, UK). Specific volume was calculated dividing bread volume by bread weight, and then it was expressed as cm^3^/g. 

Crumb texture was determined by a Texture Profile Analysis (TPA) test using a TA-XT2 texture analyzer (Stable Microsystems, Surrey, UK) with a 25-mm diameter cylindrical aluminum probe. Fifty percent of depth, a trigger force of 5 g, 15 mm compression, a test speed of 1 mm/s, and a 10 s delay between the first and the second compression were the experimental conditions. Two central slices (30 mm thick) from two pieces of each bread elaboration were analyzed. Hardness (N), springiness, resilience and cohesiveness were calculated [27]. The increase in hardness was calculated as the difference between the hardness on day 7 and at 24 h expressed as a percentage. 

An HP Scanjet G3110 scanner (HP, Palo Alto, CA, USA) was used to obtain the slice images.

#### 2.2.5. Consumer Test

A hedonic sensory evaluation of the breads was conducted with 80 volunteers from the College of Agricultural Engineering in Palencia (Spain), between 18 and 66 years old, of various socioeconomic backgrounds. Samples were analyzed one day after baking. For odor, texture, and taste evaluation, samples were presented in slices (30 mm thick slice) coded with four-digit random numbers and served in a random order. For visual appearance evaluation, a full bread of each ample was shown to the volunteers. Finally, the overall appreciation was evaluated. In the nine-point hedonic scale, the values ranged from “like extremely” to “dislike extremely”, punctuated from “9” to “1”, respectively. 

#### 2.2.6. Statistical Analysis

The results were evaluated by simple ANOVA (one-way analysis of variance) using Statgraphics Centurion XVII software (StatPoint Technologies, Warrenton, VA, USA). Fisher’s least significant difference (LSD) test was used to differentiate means at a 95% significance level (*p* < 0.05).

## 3. Results and Discussion

### 3.1. Starches and Flour Properties

As Table 2 shows, for both starch blends and starch and rice flour blends, the citrus fiber incorporation increases water-binding capacity (WBC) values. This is because the water absorption capacities of citrus fibers are higher [28] than those of starches and rice flours. Furthermore, the activation of citrus fiber by mechanical pretreatment increases these values even more. This increase in WBC with fiber activation has been attributed, in other vegetable fibers, to a breakdown of the cells and the cell walls, which leads to the generation of a three-dimensional network [29], as well as to an increase in its surface area and its interaction with water [30]. In the case of citrus fiber, this increment in WBC agrees with results observed by Schalow et al. [28] with a passage through ultraturrax—a similar treatment to that used in this study—by Su et al. [22] after high pressure homogenization, and by Jiang et al. [21] after passage through a ball mill. In all cases, the fibers were subjected to stress and particle breakage. Finally, it should be noted that this increase is greater in the case of the rice flour mixture than in the case of the starch mixture, which seems to indicate a greater interaction between the citrus fiber and the rice flour particles, which are more irregular in shape than the starches, although larger in size.

RVA curves (Figure 2) show that the flour blend has lower viscosity values throughout the curve and reaches the peak viscosity later than the non-flour blends. Sigüenza-Andrés et al. [26] already observed that rice flour presented these differences compared to corn starches and, especially, to tapioca starches. The incorporation of citric fiber increases peak viscosity and, generally, the viscosity from this point on. It is known that citrus fiber is composed mostly of cellulose and pectin, with smaller amounts of lignin and hemicelluloses [9]. Thus, the increase in the pasting properties of the blends may be due to the presence of pectins—soluble fibers with high thickening capacity—as already proved by Ma et al. [31], and celluloses, as shown by Díaz-Calderón et al. [32]. This increase is greater when citrus fiber is activated, as in the case of WBC. In the case of mixtures with flour, the effect of citrus fiber is also clearer, as in WBC. Therefore, both effects may be related to its higher water absorption capacity and thickening power.

Solid black line: Temperature; Solid grey line: Control; Dashed line: Mixture with citrus fiber; Dotted line: Mixture with activated citrus fiber.

As for gel hardness, the flourless mixture showed much higher values than the flour mixture (Table 2). These are logical values since the fraction that gelatinizes and then retrogrades the starch, which is higher in the mixtures without rice flour, compared to those with pure starch. In addition, Bravo-Nuñez et al. [25] already observed that rice proteins interfered with the rearrangement of amylose chains, reducing the strength of the gels. Citrus fiber reduces the hardness of the mixes without flour and does not modify that of the mixes with flour, regardless of whether it is activated or not. In this case, it is logical that the reduction of the total amount of starch by incorporating another ingredient in the mixture with less gelling capacity causes a reduction in the hardness of the gel, since starch is the main responsible for this gelling. Although it is true that some pectins of high and medium esterification can increase the hardness of starch gels, those of low esterification do not [33]. The strength of citrus fiber gels has been shown to be much lower than that of pectins [28].

### 3.2. Gluten-Free Bread Characteristics

To calculate the optimum hydration of each formulation, breads were made with increasing hydrations, starting from 90%. However, in the case of breads with CF, it was necessary to increase this minimum in order to form a manageable dough. The curves obtained can be seen in Figure 1. Previous works confirm that the specific volume of breads increases as hydration does until it reaches an optimum specific volume, followed by a collapse of the dough occurs because it cannot support the structure, and thus volume loss ensues [7,8,25]. This is confirmed in our case viewing the curves. Unlike wheat breads, for which equipment such as the farinograph is often used to calculate optimal hydration, in gluten-free breads there is no evidence that a single rheology guarantees an optimum. In fact, in previous works, changing the formulation modifies the optimal rheology of the doughs to achieve the highest specific volume [8]. 

The optimal hydration of control gluten-free bread formulations (Table 3) is the same for both mixtures (100%), with no differences in the specific volume between them. In both cases, the optimal hydration is clearly increased by incorporating citrus fiber. This result agrees with that observed by Korus et al. [18], who calculated the optimum hydration with a dough texture test. It also coincides with other works that analyze the incorporation of fibers with high water absorption capacity, such as psyllium [34,35]. In the case of the mixture with rice flour, the incorporation of CF, or ACF, greatly reduces the specific volume of the breads and, although the optimum is set at 160, there is an important interval where no significant differences are observed between the specific volumes of the breads. A very open and irregular porosity was also observed, especially in the case of inactivated citrus fiber (Figure 3). In this case, it was seen that after a certain hydration, the gas produced escaped from the dough, but this did not lose volume or, at least, not as abruptly as in the control. In the case of activating the fiber, the optimum hydration and the final specific volume were similar and, although an irregular crumb alveolus and horizontally elongated cells were observed, possibly indicating coalescence phenomena, this alveolus was more closed than that of the inactivated fiber. Although the emulsifying properties of citrus fiber in water/oil mixtures have been demonstrated [12], it seems that in the case of gluten-free breads, which are similar to a foam and are formed by air bubbles surrounded by a whipped dough, citrus fiber breaks this alveolar structure, promoting the coalescence of the bubbles and gas loss. This effect is enhanced when the viscosity of the mass surrounding the bubbles is lower [13], and therefore, hydration increases. Even when the higher values of G′ and G′′ (Table 2) of the ACF mixtures seem to minimize this problem, the volume loss remains unsolved. An increase of these values and a decrease of Tan delta of the doughs was observed with the addition of CF, especially ACF, when the optimum hydration was reached. However, as shown in Figure 1, it is possible to increase hydration, and therefore reduce G′ and G′′ values, without a large volume loss, in contrast to other formulations.

Data are expressed as mean ± Standard Deviation of duplicate trials. Values with the same letter do not show significant differences (*p* < 0.05).

In a mixture consisting exclusively of starches, mostly corn starch, with smaller particle size, and thus with a greater capacity to stabilize foams and generate products with a more closed alveolus, partly thanks to the Pickering effect [3], the effect of CF is much lower. The more open alveolus noticed in breads with CF confirms the negative effect on the stabilization of the foams created by CF. However, this does not become “dangerous”, and the structure does not collapse until a certain hydration is reached. At this point, the viscosity of the dough surrounding the bubbles decreases and stops supporting the weight of the structure. A significant specific volume drop is seen with optimal hydration when incorporating CF and, to a greater extent, ACF. This agrees with Korus et al. [18], who also used a formulation with corn starch and a tuber (in their case potato) and also observed a similar effect on the alveolus. However, in their work, the specific volume of the breads is much smaller than in ours. This ratifies the importance of the formulation, since in their work, HPMC was not used; instead, a mixture of gums (pectins and guar) as gluten substitutes were used. In fact, HPMC has a very different effect on gluten-free breads compared to gums, greatly improving the volume of the breads [5], but being more sensitive to differences in hydration. As in the rice flour mixture, an increase in optimal hydration is observed; this is so because, in the case of ACF, it has higher WBC and thickening power, and thus requires higher hydration to equalize the rheology of the doughs. A good shape is also observed, with a higher central part, which denotes that there has been no fall there, and although the alveolus is somewhat coarser than that of the non-fiber mixture, it is rounded, as in the control. In starch mixtures, unlike rice flour mixtures, the rheology of the doughs with optimum hydration is similar in all of them, while higher values of G′ and G′′ are only observed in the case of CF.

Another important point to highlight is that, when CF is incorporated, the amount of water in the formulas increases, and yet no greater loss of moisture is observed during baking; in the case of mixture with starches, it is even reduced with ACF. This is due to the high water absorption capacity of the fibers already discussed, which seems to be preserved after baking. Belorio and Gómez [5] reported a lower weight loss in breads made with hydrocolloids, with high water absorption capacity, versus those made only with HPMC, with lower water absorption capacity. However, the lower moisture loss may also be due to the smaller volume of the breads, and thus their lower surface and exchange area [7]. Therefore, breads with CF and ACF, with much higher hydrations, will exhibit higher moisture, and thus juiciness. This is important because one of the typical problems of gluten-free breads is their drier texture [6,36], something that CF and ACF may solve, at least in part. However, this higher moisture may also reduce bread shelf life due to microbiological problems brought about by the higher water activity of the breads, an aspect that needs to be studied.

Regarding the crumb texture of the breads (Table 4), it can be seen that the addition of CF or ACF increased hardness on breads made with starch and flour mixtures. This effect may be due to the lower specific volume of breads with fiber, something that is in keeping with results observed in various studies [7,26,37]. However, in the case of starch mixtures, despite the lower specific volume of breads with fiber, no significant differences in hardness were observed. In these cases, it seems that the negative effect of the specific volume may be compensated by the lower gel strength of the mixtures with fiber, or by their lower final viscosity values in the RVA curves, already commented. It is also important to note that the evolution of hardness is much lower in the case of breads with CF and ACF, in both formulations, a result that agrees with those observed by Korus et al. [18]. This point is very important, since rapid staling is one of the main problems of gluten-free breads compared to those made with wheat flour. These differences are mainly due to the higher mobility of water in gluten-free breads [38], and therefore, to the loss of this water in storage. Hence, the reason for this reduction in hardness is related to the high water holding capacity of CF, and thus the maintenance of a “juicier” and less hard texture. This effect coincides with that observed in other studies that incorporated products with high water absorption capacity [39].

It should also be noted that the addition of CF or ACF allows for increased cohesiveness, resilience, and elasticity in both formulations, although in general, this effect is greater with CF. Matos and Rosell [40] have already related these parameters to the hydration properties of crumbs. Therefore, this effect may be due to the higher water retention capacity of citrus fiber.

As for the acceptability test (Table 5), breads with a mixture of starches and rice flour and fiber were not included in this test due to their low specific volume. In all the parameters analyzed, with the exception of odor, the breads with rice flour obtained worse evaluations than the breads made with starch mixtures. The addition of fiber did not change visual appearance ratings, despite the lower volume and coarser crumb alveolus, nor did it change the taste ones. However, tasters found a slightly different aroma in breads with fiber, which reduced the rating on this point. In contrast, the fiber breads showed better texture ratings. This may be related to the fact that gluten-free breads stand out negatively for the sensation of dryness, hardness, and crumbliness (easy to break in the mouth), generating a grittier sensation in contrast to wheat breads [36] The lower hardness, higher moisture, and higher cohesiveness of fiber breads can minimize these negative effects. The values obtained in this work are similar or even higher than other previous works with gluten-free breads obtained with a similar scale and number of evaluators [41].

## 4. Conclusions

The incorporation of CF increases the hydration needs of gluten-free formulas, improving dough yield. However, the effect on bread characteristics changes depending on the formula used. In a starch-based formula, it can reduce hardness and staling, improving cohesiveness, resilience, and texture perception without reducing overall acceptability. Fiber activation by shear forces further increases hydration needs and dough yield but may slightly change some of these characteristics. Therefore, in this work we show that some aspects related to these fibers, such as formulation, pretreatment, and probably the way the ingredients are mixed, have a great influence on the results obtained. Moreover, with the addition of an adequate amount of citrus fiber, the nutritional claims “source of fiber” (3 g/100 g or 1.5 g/100 kcal) or “high fiber” (6 g/100 g or 3 g/100 calories) could be used. In addition, the incorporation of CF improves the perception of the texture by consumers and does not reduce the overall acceptability, and even improves it (flour and starch-based formula).

## Figures and Tables

**Figure 1 foods-12-01357-f001:**
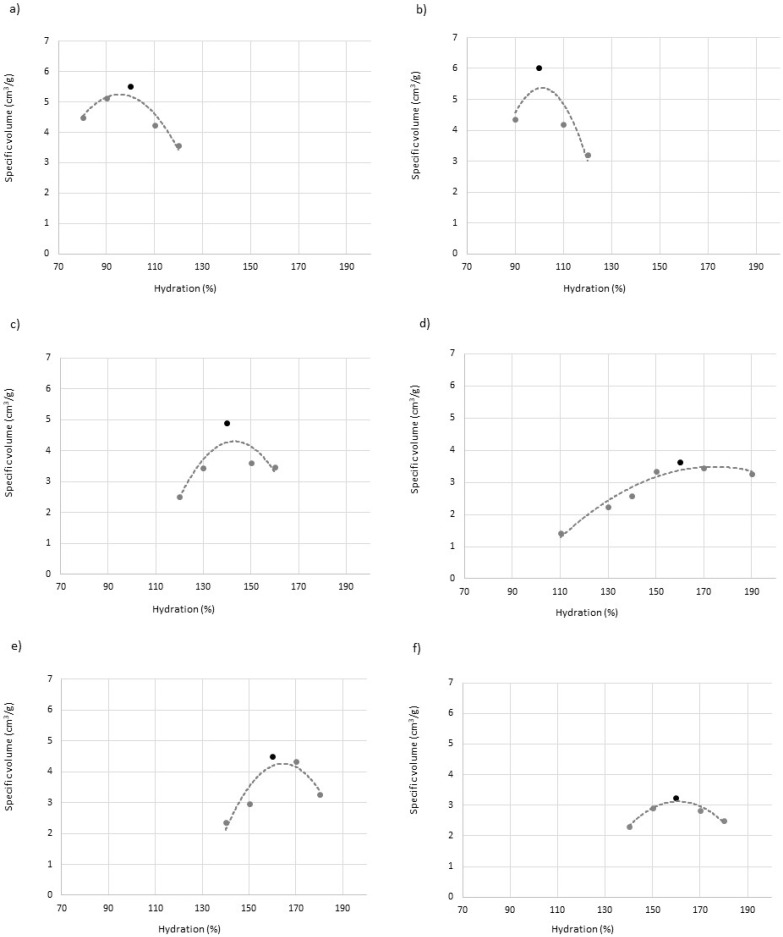
Hydration versus specific volume of gluten-free breads. Control 1 (**a**); Control 2 (**b**); CF 1 (**c**); CF 2 (**d**); ACF 1 (**e**); ACF 2 (**f**). Black dots indicate selected optimal hydration. CF: Citrus Fiber; ACF: Active Citrus Fiber. 1: Starch-based formula; 2: Flour and starch-based formula.

**Figure 2 foods-12-01357-f002:**
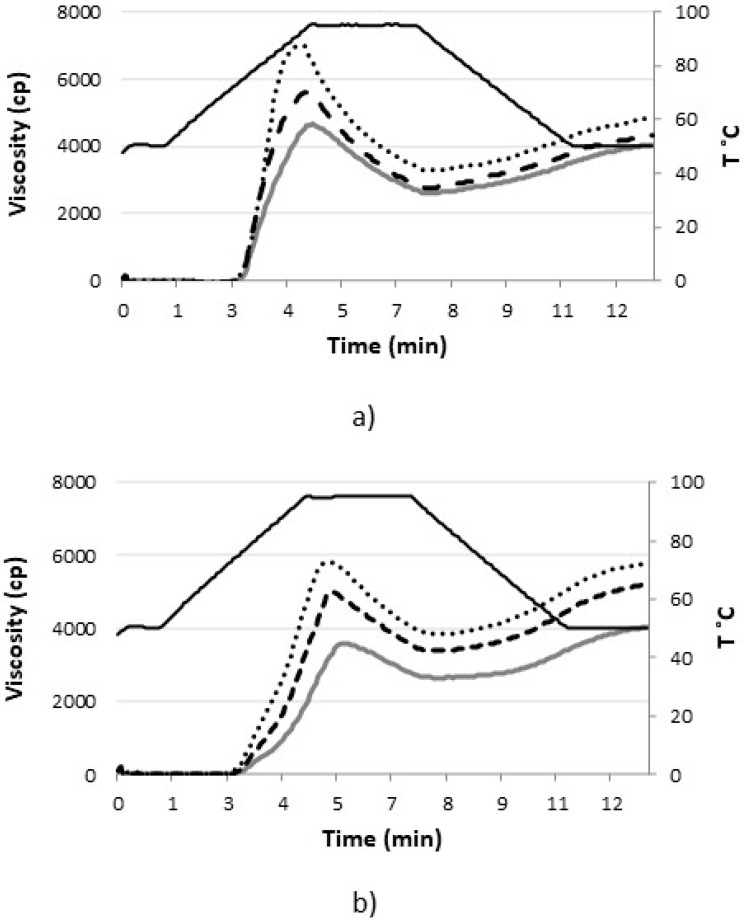
Pasting properties of flour/starches/fiber mixtures. (**a**) 40% maize starch/40% Rice Flour/20% Cassava Starch. (**b**) 80% maize starch/20% Cassava Starch.

**Figure 3 foods-12-01357-f003:**
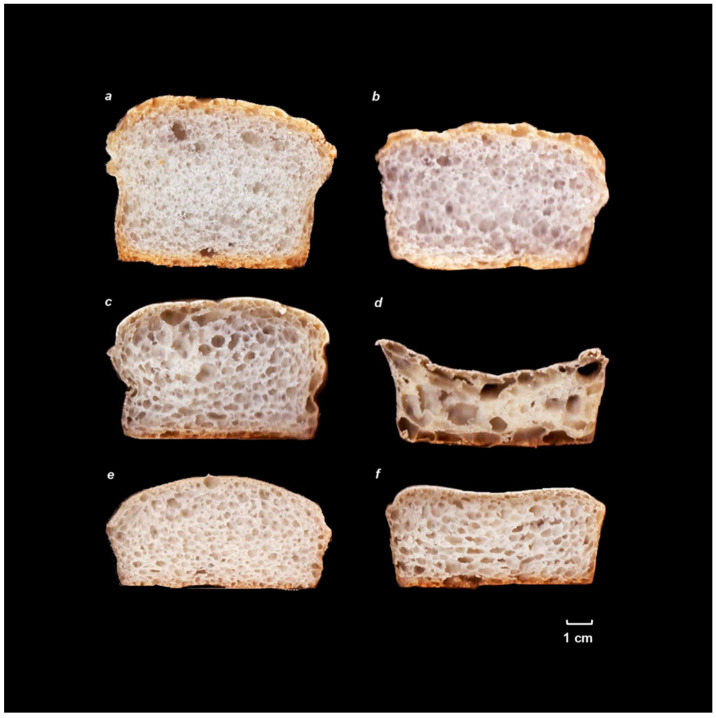
Images of gluten free breads. Control 1 (**a**); Control 2 (**b**); CF 1 (**c**); CF 2 (**d**); ACF 1 (**e**); ACF 2 (**f**).

**Table 1 foods-12-01357-t001:** Formulation (in grams) of bread samples prior to the determination of their optimum moisture content.

Ingredients	Control 1	CF 1	ACF 1	Control 2	CF 2	ACF 2
Water	x	x	x	x	x	x
Rice flour	-	-	-	160	160	160
Maize starch	320	320	320	160	160	160
Cassava starch	80	80	80	80	80	80
Sunflower oil	24	24	24	24	24	24
White sugar	20	20	20	20	20	20
Yeast	12	12	12	12	12	12
Salt	8	8	8	8	8	8
HPMC	8	8	8	8	8	8
CF	-	24	-	-	24	-
ACF	-	-	24	-	-	24

HPMC: Hydroxypropyl Methyl Cellulose; CF: Citrus Fiber; ACF: Active Citrus Fiber.

**Table 2 foods-12-01357-t002:** Properties of flour/starches/fiber mixtures.

	WBC (Water g/Solid g)	Gel Hardness (N)	G′ (Pa)	G″ (Pa)	tanδ
Control 1	0.85 ± 0.01 a	21.58 ± 0.53 c	44.90 ± 5.01 a	81.29 ± 7.12 a	1.88 ± 0.01 c
CF 1	1.54 ± 0.03 b	14.97 ± 2.51 b	227.50 ±1.56 b	253.15 ± 2.05 b	1.16 ± 0.01 b
ACF 1	2.87 ± 0.28 d	11.97 ± 0.45 ab	56.39 ± 1.30 a	103.60 ± 2.26 a	1.97 ± 0.10 c
Control 2	1.01 ± 0.03 a	8.60 ± 3.11 a	181.25 ± 4.38 b	203.68 ± 8.17 b	1.25 ± 0.06 b
CF 2	2.15 ± 0.09 c	9.38 ± 3.19 a	323.40 ± 88.53 c	350.55 ± 61.87 c	1.15 ± 0.11 b
ACF 2	4.44 ± 0.28 e	7.92 ± 0.35 a	793.10 ± 15.27 d	603.70 ± 11.31 d	0.78 ± 0.01 a

WBC: Water-Binding Capacity; CF: Citrus Fiber; ACF: Active Citrus Fiber. 1: Starch-based formula; 2: Flour and starch-based formula. Data are expressed as mean ± Standard Deviation of duplicate trials. Values with the same letter do not show significant differences (*p* < 0.05).

**Table 3 foods-12-01357-t003:** Specific volume and weight loss of gluten-free breads with or without CF.

	Optimal Hydration (%)	Specific Volume (cm^3^/g)	Weight Loss (%)
Control 1	100	6.15 ± 0.44 c	26.38 ± 1.94 bc
CF 1	140	4.65 ± 0.13 b	23.48 ± 2.44 ab
ACF 1	180	3.72 ± 0.14 a	20.69 ± 1.32 a
Control 2	100	6.01 ± 0.23 c	28.40 ± 2.34 c
CF 2	160	3.51 ± 0.01 a	26.82 ± 0.32 bc
ACF 2	160	3.22 ± 0.15 a	26.76 ± 0.59 bc

CF: Citrus Fiber; ACF: Active Citrus Fiber, 1: Starch-based formula; 2: Flour and starch-based formula. Data are expressed as mean ± Standard Deviation of duplicate trials. Values with the same letter do not show significant differences (*p* < 0.05).

**Table 4 foods-12-01357-t004:** Textural parameters of gluten-free breads with or without CF.

	Hardness (N)	Cohesiveness	Resilience	Elasticity (N)	Hardness Increase (%)
Control 1	4.86 ± 0.06 bc	0.31 ± 0.0052 a	0.11 ± 0.0011 b	0.98 ± 0.01 a	227.23 ± 12.58 b
CF 1	4.25 ± 0.05 b	0.49 ± 0.0071 d	0.24 ± 0.0025 e	2.44 ± 0.04 c	73.62 ± 4.89 a
ACF 1	5.26 ± 0.03 c	0.49 ± 0.0071 d	0.23 ± 0.0026 d	1.69 ± 0.09 b	50.83 ± 51.14 a
Control 2	3.31 ± 0.05 a	0.30 ± 0.0061 a	0.10 ± 0.0046 a	0.96 ± 0.00 a	415.49 ± 17.20 c
CF 2	6.87 ± 0.13 d	0.43 ± 0.0018 c	0.12 ± 0.0006 bc	2.69 ± 0.11 d	82.07 ± 20.85 a
ACF 2	7.42 ± 0.62 d	0.33 ± 0.0081 b	0.12 ± 0.0018 c	1.63 ± 0.01 b	52.80 ± 21.84 a

CF: Citrus Fiber; ACF: Active Citrus Fiber; 1: Starch-based formula; 2: Flour and starch-based formula. Data are expressed as mean ± Standard Deviation of duplicate trials. Values with the same letter do not show significant differences (*p* < 0.05).

**Table 5 foods-12-01357-t005:** Acceptability of gluten-free breads with or without CF.

	Visual Appearance	Odor	Texture	Taste	Overall Acceptability
Control 1	6.72 ± 1.55 b	6.63 ± 1.57 b	6.20 ± 1.63 b	6.41 ± 1.66 b	6.50 ± 1.25 b
Control 2	5.29 ± 1.78 a	6.26 ± 1.46 ab	4.99 ± 1.85 a	5.58 ± 1.78 a	5.51 ± 1.58 a
CF 1	6.77 ± 1.28 b	5.90 ± 1.51 a	6.73 ± 1.38 c	6.29 ± 1.79 b	6.56 ± 1.23 b
ACF 1	7.11 ± 1.41 b	5.84 ± 1.72 a	6.76 ± 1.49 c	6.21 ± 1.79 b	6.72 ± 1.36 b

CF: Citrus Fiber; ACF: Active Citrus Fiber; 1: Starch-based formula; 2: Flour and starch-based formula. Data are expressed as mean ± Standard Deviation of duplicate trials. Values with the same letter do not show significant differences (*p* < 0.05).

## Data Availability

The data presented in this study are available on request from the corresponding author.

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
