# Peer review of "Can Citrus Fiber Improve the Quality of Gluten-Free Breads?"

_foods, 2023, doi:10.3390/foods12071357_

Round 1

Reviewer 1 Report

General comments:

1.     Line 122: why was the texture analyzed only after 1 and 7 days? Why not after 2 and 3 days as bread is usually consumed quickly.

2.     The sensory evaluation should be performed in a group of people on a gluten-free diet as the scores for gluten-free formulations

Detailed comments:

3.     Line 11: please delete a full stop.

4.     Line 45: please reformulate ‘large portions’.

5.     Line 29: allergy to what? Please correct the sentence.

6.     Line 30: please change into ‘pseudo-cereals and gluten-free grains’.

7.     Line 32: it is not a necessity; however, indeed, is often used. Please reformulate the sentence.

8.     Line 37: please delete ‘among others’.

9.     Lines 42-43: the sentence ‘ Unlike wheat flour breads, in gluten-free breads no 42 equipment indicates the optimal hydration.’ Is not clear. Please clarify.

10.  Line 79: please provide more details for Citrus Fiber.

11.  Line 85: please correct the formatting.

12.  Line 107: please add the reference for the bread recipe.

13.  Lines 119-122: Materal & methods section should be clearly separated from the Results section (Figure 1 applies to results).

14.  Figures 1 and 2: the graphs should be corrected to more attractive (e.g., te font should be unified, the font size should be adjusted).

15.  Table 1: please add units.

16.  Line 161: please correct the unit typo.

17.  Table 5: please change ‘global acceptability’ into ‘general acceptability’

18.  Lines 359-369: please refer to the literature data on the acceptability of gluten-free breads. Conclude if the received scores are acceptable or not.

19.  Conclusions: the comment on the sensory evaluation should be added.

Author Response

First of all I would like to thank the reviewer for his work and for the useful comments made.

Comments and Suggestions for Authors

General comments:

  1. Line 122: why was the texture analyzed only after 1 and 7 days? Why not after 2 and 3 days as bread is usually consumed quickly.

The days of measurement is something that we have considered many times, and we normally use days 1 and 7 (we already have several published articles analyzing the texture on these days). There are many reasons:

On the one hand, due to the equipment to prepare the doughs that we have, in one preparation we only obtain enough loaves to analyze the bread for two days, if we want to measure at least two loaves each day. We can make more elaborations with the same formulation to measure more days, but they would no longer correspond to the same elaboration. We only do this when the article is very focused on improving its shelf life by reducing staling. In this case analyse the texture changes has been considered important, but the article has not focused solely on that aspect

We believe that day 1 is necessary because it indicates the initial texture at a logical moment of consumption, especially if it is produced in a small bakery.

If the elaboration is more industrial, a slightly longer shelf life is sought (as long as possible). We know that the texture of the breads changes very quickly in the first days and then the changes slow down. We believe that after 7 days we already see a texture that gives us an idea of this point, something that we do not believe will be fully achieved on days 2 and 3, at least in some elaborations, such as those of the fiber-enriched breads of this work, which, as can be seen, increase the hardness much less than control.

  1. The sensory evaluation should be performed in a group of people on a gluten-free diet as the scores for gluten-free formulations

We believe that the most important thing is that the number of consumers is very high, with a minimum of 70. Otherwise, the acceptability score is not valid, since different people have different preferences and can affect the results. Getting a large number of consumers and organizing these tests is complicated, due to people's time availability and the need for the loaves to be tasted with the same storage time. Trying to do it with people with celiac disease or those who follow a gluten-free diet would be really dificult. People who follow a gluten-free diet participate in our tastings and we take their opinions into account, which normally coincide with those of the majority, although they value research into this type of product somewhat more, but since it is not as numerous as we believe necessary to draw valid conclusions we do not include them

On the other hand, the success of many of these products is that their consumption is not only limited to people with celiac disease or who follow a strict gluten-free diet, but also to other people who consider these products healthier, but do not consume them for its lack of quality and high price, family members of celiacs, and other unrestricted consumers. For this reason, we believe that it is appropriate to carry out tastings with consumers without restrictions.

However, we believe that it would be very interesting to carry out studies comparing the opinions of different groups in order to have more information on these aspects. We do not know them.

 Detailed comments:

  1. Line 11: please delete a full stop.

We do not know if there is an error in the indicated line number. In line 11 there is only one point and it separates two independent sentences. We do not think it makes sense to remove it.

  1. Line 45: please reformulate ‘large portions’.

We think there is also an error and the reviewer refers to line 24. It has been changed. We appreciate the comment.

  1. Line 29: allergy to what? Please correct the sentence.

We were referring to wheat allergy, since we were talking about people who cannot eat wheat. We have clarified it.

  1. Line 30: please change into ‘pseudo-cereals and gluten-free grains’.

Done

  1. Line 32: it is not a necessity; however, indeed, is often used. Please reformulate the sentence.

It has been modified taking into account the indications of the reviewer

  1. Line 37: please delete ‘among others’.

Done

  1. Lines 42-43: the sentence ‘ Unlike wheat flour breads, in gluten-free breads no 42 equipment indicates the optimal hydration.’ Is not clear. Please clarify.

The sentence has been modified and is as follows “Unlike wheat flour breads, in gluten-free breads there is no universally accepted equipment to analyze the hydration of the dough and modify it based on changes in the recipe, such as the farinograph.”

  1. Line 79: please provide more details for Citrus Fiber.

We have incorporated the trade name. Although we have the technical fiber of the product, this does not give specific information, but maximums and minimums. We do not have more information. We believe that with the indication of the brand and trade name the experiment can be replicated. In addition, we have included not only the analyzes of the breads, but also the analysis of the WBC, RVA and gel properties of the mixtures, something that gives an idea of the properties of citrus fiber.

  1. Line 85: please correct the formatting.

It has been changed

  1. Line 107: please add the reference for the bread recipe.

It has been added

  1. Lines 119-122: Materal & methods section should be clearly separated from the Results section (Figure 1 applies to results).

We agree with the reviewer on the separation of materials and methods in the results sections. However, according to the indications of the publication, the figures and tables must be placed close to the text where they are cited for the first time. And figure 1 is cited in the materials and methods section, since it is necessary to understand how the optimal hydration with which the different breads have been made has been calculated.

  1. Figures 1 and 2: the graphs should be corrected to more attractive (e.g., te font should be unified, the font size should be adjusted).

They have been changed

  1. Table 1: please add units.

They have been added

  1. Line 161: please correct the unit typo.

Done

  1. Table 5: please change ‘global acceptability’ into ‘general acceptability’

“global acceptability has been changed into “overall acceptability” as reviewer 3 suggested

  1. Lines 359-369: please refer to the literature data on the acceptability of gluten-free breads. Conclude if the received scores are acceptable or not.

We appreciate the comment. It is something that we have considered, but we have decided not to do it. From our experience it is very difficult to compare data from different acceptability tests. The values depend a lot on the scale used, the number of tasters, the samples presented, the time that elapses from the preparation to the tasting, etc. In our case, all the values are obtained in the same session, and we have verified that there are significant differences with those of another session, or when the samples presented are different (tasters unconsciously compare between the evaluated samples, and the presence of better or worse samples conditions the score of some that can be repeated in different tastings). The values obtained in this work are similar or even higher than other previous works with gluten-free breads obtained with a similar scale and number of evaluators (Steffolani et al., 2014). In addition, our group has studied the world's commercial breads and the formula used is based on that study and the most widely used ingredients worldwide (Román et al., 2019). We have also advised different companies in the development of gluten-free breads and other products that are on the market, with formulations based on the one used in this work, modifying it to obtain different industry requirements. Since the object of the study is not to evaluate the control formulation, but rather the effect of incorporating CF on it, and to see that it cannot only improve other aspects, such as texture and shelf life, but also that it has no negative effects (many times undetected through instrumental analysis, such as taste or smell) on sensory properties, I think we can affirm, based on the results, that there is indeed no negative effect on the inclusion of citrus fiber, or it can even improve acceptability.

Roman, L., Belorio, M., Gomez, M. (2019) Gluten-free breads: The gap between research and commercial reality. Comprehensive Reviews in Food Science and Food Safety, 18:690-702. DOI:10.1111/1541-4337.12437

Steffolani, E., De la Hera, E., Perez, G., Gómez, M. (2014) Effect of chia (Salvia hispanica L) addition on the quality of gluten-free bread. Journal of Food Quality, 37:309-317. DOI:10.1111/jfq.12098.

  1. Conclusions: the comment on the sensory evaluation should be added.

It has been added.

Reviewer 2 Report

Article Type: Article
Title:
Can citrus fiber improves the quality of gluten-free breads?

Journal: Foods-2246007

It is of crucial importance to improve the nutritional and technological quality of gluten-free bread and the use of normal and shear-activated citrus fiber can be an interesting approach. In gluten-free doughs, the understanding of the hydration impact on volume, rheology, and on bread quality is an important topic. Besides rheology and sensory data, it would be interesting to compare the nutritional value of the studied formulations enriched in citrus fibre, and check the possibility of using nutritional claims.

The article contains new information, the aim of the work is clearly established, the experiments well conducted, discussion and conclusions are well documented, despite some concerns.

L49: Replace “citrus fiber” by “citrus fiber (CF)”.

L60: Please correct, “mill” instead of “mil”.

L89: The proportions used in the blends (rice flour, starches and CF) should be indicated.

L98-104: Which was the distance or % compression used in the texture tests?

Figure 1: Please indicate the meanings of CF, ACF, 1 and 2. Replace cm3 by cm3.

L153-154: Why not measuring texture after cooling, on the first day? It would be nice to compare those results with 24h after baking.

L188: Please use replace “WBC” by “water binding capacity (WBC)”.

L198-200: particle size of starches and rice flour should be determined since it greatly influences the results.

Table 2: There is no caption. Change commas to points (e.g. 0.85 instead of 0,85). Values of G´ and G´´ were taken at which frequency? G´ and the G´´ unit is missing. The optimal hydration used for each formulation should be indicated in this table.

Figure 2: The optimal hydration used for each formulation should be indicated also here. CP is not an SI unit, please change it. There are no colours in the graphs, despite colours are mentioned in the caption.

L233-239: I do not totally agree with this comment since the authors did not replace flours by CF. Please revise/adjust it.

L275: Please correct, “doughs” instead of “masses”.

Table 3: Change commas to points.

Table 4: Change commas to points. Explain the meaning of Hardness increase (%).

Table 5: Change commas to points.

Author Response

First of all I would like to thank the reviewer for his work and for the useful comments made.

It is of crucial importance to improve the nutritional and technological quality of gluten-free bread and the use of normal and shear-activated citrus fiber can be an interesting approach. In gluten-free doughs, the understanding of the hydration impact on volume, rheology, and on bread quality is an important topic. Besides rheology and sensory data, it would be interesting to compare the nutritional value of the studied formulations enriched in citrus fibre, and check the possibility of using nutritional claims.

The article does not intend to improve the nutritional quality of gluten-free breads, nor to reach a certain level of fiber, but rather its technological quality and its shelf life. For this reason we have not measured the nutritional values. Obviously, by incorporating fiber, the fiber content became, but as I said, it was not the objective of the work. We have included a sentence in the conclusions indicating that by regulating the amount of added fiber, nutritional claims could be used.

The article contains new information, the aim of the work is clearly established, the experiments well conducted, discussion and conclusions are well documented, despite some concerns.

L49: Replace “citrus fiber” by “citrus fiber (CF)”.

Done

L60: Please correct, “mill” instead of “mil”.

Done

L89: The proportions used in the blends (rice flour, starches and CF) should be indicated.

We appreciate the comment. They are the same proportions that are used in bread formulations, and are shown in Table 1, but excluding the rest of the ingredients. It has been indicated in the text.

L98-104: Which was the distance or % compression used in the texture tests?

Thanks for the indication, the information has been included (15mm compression)

Figure 1: Please indicate the meanings of CF, ACF, 1 and 2. Replace cm3 by cm3.

The meanings have been added, and the units changed.

L153-154: Why not measuring texture after cooling, on the first day? It would be nice to compare those results with 24h after baking.

It is a good appreciation and we have ever considered it. The reason is double. In general, these breads are not consumed within a few hours of being made, like other types of wheat bread, but after a few days. On the other hand, we know that texture changes in the first hours are faster and the time that elapses from the time they come out of the oven until the measurements are made is very important, so differences of a few minutes can translate into differences in the data obtained, even between different repetitions, and it is not always possible to guarantee this accuracy. After 24 hours the texture is more stable and guarantees more reproducible results. For these reasons we prefer to analyze it at 24 hours and 7 days.

In any case, data correctly obtained a few hours after baking would also be useful.

L188: Please use replace “WBC” by “water binding capacity (WBC)”.

Done

L198-200: particle size of starches and rice flour should be determined since it greatly influences the results.

The particle size of starches is quite stable, depending on the origin of the starch, so we have not measured it. Corn starch and tapioca starch are very fine, between 5-20 microns. It is true that rice flours can present a greater dispersion and influence the final result, especially in formulations that incorporate them in higher amounts. We have included the particle size of this flour, as stated by the reviewer, for this reason. In our case, the flour had an average particle size of 136 microns.

Table 2: There is no caption. Change commas to points (e.g. 0.85 instead of 0,85). Values of G´ and G´´ were taken at which frequency? G´ and the G´´ unit is missing. The optimal hydration used for each formulation should be indicated in this table.

Thanks for the comments. The units of G' and G have been included. Commas have been changed to points. A table title has been included.

It is difficult to incorporate the optimal hydration in the table since this only influences the rheology measurements, not the WBC and gel hardness measurements. To facilitate the understanding of the reader, the indication of optimal hydration has been included in Figure 1, so that it appears in the article before Table 2 and the reader can easily access this information.

The frequency (1 Hz) used to obtain the values of G' and G" has been included in materials and methods.

Figure 2: The optimal hydration used for each formulation should be indicated also here. CP is not an SI unit, please change it. There are no colours in the graphs, despite colours are mentioned in the caption.

The RVA method is a well-known official method in the field of starch and flour analysis. In it the amount of water is fixed (4g of flour mixed with 25 ml of water). Therefore, hydration is not modified, and we consider that it is not necessary to indicate it, by following an official referenced method. It is true, as the reviewer says, that cp is not an SI unit. However, the results of this analysis are always given in this type of units, which are the ones indicated by the team, and we believe that for a better understanding by readers accustomed to this methodology, it is preferable to leave this unit. On the subject of colors we believe there is some error. We only indicate lines in black and grey colors or tones that are seen in the graphs, and are distinguished from the lines that are also in black, but discontinuous. We have incorporated the term solid black to make the difference clearer.

L233-239: I do not totally agree with this comment since the authors did not replace flours by CF. Please revise/adjust it.

We appreciate the comment, and it is true that there has not been a substitution, but by incorporating the CF the amount of starch is reduced in the mixture. The wording has been corrected to express it correctly.

L275: Please correct, “doughs” instead of “masses”.

Done

Table 3: Change commas to points.

Done

Table 4: Change commas to points. Explain the meaning of Hardness increase (%).

Done. The meaning of increased hardness has been incorporated into materials and methods section.

Table 5: Change commas to points.

Done

Reviewer 3 Report

The main objective of this work is to investigate the effect of citrus fiber (without or after shear forces activation) on the properties of gluten-free bread. This is an exciting study, and the authors have collected a sufficient dataset using a suitable methodology. The manuscript is generally well-written and structured. Results are well-presented and discussed. However, it needs revision to be accepted for publication. No proximate and nutritional composition was evaluated for the final products. Other comments are listed below.

1.      Abstract - Please include a brief conclusion in the abstract.

2.      Provide the RVA profiles for the determination of pasting properties.

3.      Include the brand of the oven used in this study.

4.      Include the dimension of the bread used for sensory evaluations.

5.      Why was colour measurement not conducted in this study?

6.      Please use overall acceptability instead of global acceptability.

7.      To provide better sensory scores, it is better to compare the treatments with commercial gluten-free bread!

Author Response

First of all I would like to thank the reviewer for his work and for the useful comments made.

Comments and Suggestions for Authors

The main objective of this work is to investigate the effect of citrus fiber (without or after shear forces activation) on the properties of gluten-free bread. This is an exciting study, and the authors have collected a sufficient dataset using a suitable methodology. The manuscript is generally well-written and structured. Results are well-presented and discussed. However, it needs revision to be accepted for publication. No proximate and nutritional composition was evaluated for the final products. Other comments are listed below.

  1. Abstract - Please include a brief conclusion in the abstract.

We had not included a brief conclusion due to the limitations of the journal, which indicates that it should not exceed 200 words, and due to the difficulty of summarizing all the parameters studied. To include this brief conclusion, we have removed an initial sentence where we justified the research, and we have highlighted the most interesting effects.

  1. Provide the RVA profiles for the determination of pasting properties.

The RVA profile can be found in the referenced method, which is an official method. However, it can also be seen in Figure 2 (the black line indicates the temperature profile throughout the test). We believe that this information is sufficient and the reader has all the necessary information to interpret the analysis.

  1. Include the brand of the oven used in this study.

The brand of the oven has been included (Salva, Lezo, Spain)

  1. Include the dimension of the bread used for sensory evaluations.

The volunteers had a whole loaf to analyze the visual appearance, as indicated in materials and methods. For the rest of the elaborations, they received a 30mm thick slice. This information has been included in the materials and methods section.

  1. Why was colour measurement not conducted in this study?

In our studies we usually include the analysis of the color of the crumb and crust. However, we only include them in the articles when they are relevant. In this case, no significant differences have been found between the color of the crumb and crust when citrus fiber was added, something that was to be expected due to the whitish color of the fiber, which should not modify the color of the crumb and for not greatly affect the Maillard reaction and caramelization of the crust. The possible color differences that can be seen in Figure 3 are due to an optical effect due to the different alveolate and are not appreciated when the color is analysed. We have decided not to include this information so as not to lengthen the article unnecessarily.

  1. Please use overall acceptability instead of global acceptability.

It has been changed.

  1. To provide better sensory scores, it is better to compare the treatments with commercial gluten-free bread!

We appreciate the comment and it is something that we have taken into account. In this case, the object of the study is to analyze the effect of the inclusion of citrus fiber, not the development of a commercial product. In this case, we believe that it does not make much sense to compare it with a commercial bread. This comparison has several drawbacks. On the one hand, the formulation or method of preparation of commercial breads is not known, something that would be convenient to know what we are comparing. On the other hand, there are many commercial breads, with very different characteristics, and the result would depend a lot on the selected bread. Finally, it is not convenient to saturate consumers with a large number of samples (in this type of tasting it is not good to include more than 5 or 6 samples at most). Although it is true that some commercial bread could have been included, due to what was previously discussed we have decided not to. The group has experience in developing commercial gluten-free products by collaborating with companies. In those cases, there is usually a specific objective and some bread that we must surpass in acceptability. In those cases, we do incorporate this type of bread in the tastings.

Round 2

Reviewer 1 Report

The manuscript has been revised however the additional correction (#1) must be included in the text before the manuscript is accepted.

1.      (Comment #18). It is strongly suggested to include a comment on the sensory results as it was indicated in the previous review. Please expand the discussion on scores received in relation to the literature data.

2.      For the future studies on bakery products (especially bread) please conducts the study within 1, 2, 3 and 7 days.

3.      It is not true that notes received by gluten-free bread would be similar for people eating and avoiding gluten. Different studies (on different food products) show that sensory scores differ a lot between these groups. Therefore, in further studies these two groups must be separated.

Author Response

The manuscript has been revised however the additional correction (#1) must be included in the text before the manuscript is accepted.

  1. (Comment #18). It is strongly suggested to include a comment on the sensory results as it was indicated in the previous review. Please expand the discussion on scores received in relation to the literature data.

We have expanded the discussion and we have compared the data with those of another work that we have found comparable, due to the scale used and the number of tasters.

  1. For the future studies on bakery products (especially bread) please conducts the study within 1, 2, 3 and 7 days.

Thanks for the advice, we'll keep it in mind.

  1. It is not true that notes received by gluten-free bread would be similar for people eating and avoiding gluten. Different studies (on different food products) show that sensory scores differ a lot between these groups. Therefore, in further studies these two groups must be separated.

Thanks for the advice.